# Antimicrobial Peptides: Challenging Journey to the Pharmaceutical, Biomedical, and Cosmeceutical Use

**DOI:** 10.3390/ijms24109031

**Published:** 2023-05-20

**Authors:** Anna Mazurkiewicz-Pisarek, Joanna Baran, Tomasz Ciach

**Affiliations:** 1Centre for Advanced Materials and Technologies CEZAMAT, Warsaw University of Technology, Poleczki 19, 02-822 Warsaw, Poland; joanna.baran.dokt@pw.edu.pl (J.B.); tomasz.ciach@pw.edu.pl (T.C.); 2Faculty of Chemical and Process Engineering, Warsaw University of Technology, Warynskiego 1, 00-645 Warsaw, Poland

**Keywords:** antimicrobial peptides, biofilm disruption, drug development, healthcare, immunomodulatory effects, multidrug-resistant pathogens, peptide design and engineering, resistance mechanisms, smart ageing

## Abstract

Antimicrobial peptides (AMPs), or host defence peptides, are short proteins in various life forms. Here we discuss AMPs, which may become a promising substitute or adjuvant in pharmaceutical, biomedical, and cosmeceutical uses. Their pharmacological potential has been investigated intensively, especially as antibacterial and antifungal drugs and as promising antiviral and anticancer agents. AMPs exhibit many properties, and some of these have attracted the attention of the cosmetic industry. AMPs are being developed as novel antibiotics to combat multidrug-resistant pathogens and as potential treatments for various diseases, including cancer, inflammatory disorders, and viral infections. In biomedicine, AMPs are being developed as wound-healing agents because they promote cell growth and tissue repair. The immunomodulatory effects of AMPs could be helpful in the treatment of autoimmune diseases. In the cosmeceutical industry, AMPs are being investigated as potential ingredients in skincare products due to their antioxidant properties (anti-ageing effects) and antibacterial activity, which allows the killing of bacteria that contribute to acne and other skin conditions. The promising benefits of AMPs make them a thrilling area of research, and studies are underway to overcome obstacles and fully harness their therapeutic potential. This review presents the structure, mechanisms of action, possible applications, production methods, and market for AMPs.

## 1. Introduction

The discovery of antibiotics was probably one of the greatest achievements of medical sciences. During the last half of the century, antibiotics have found widespread use not only in human medicine but also in veterinary medicine and the prevention of diseases in animals. When Pasteur discovered the effect of substances produced by Penicillium moulds on bacteria, there were probably already existing counter defence mechanisms in nature, today called antibiotic resistance, which is the widespread antibiotic-induced resistance driven by the evolutionary pressure acting on various bacteria species. Antimicrobial resistance (AMR) is now on track to become the leading cause of world death in the coming decades. In 2019, almost 5 million deaths were associated with AMR, of which 1.3 million were directly attributable to resistant infections [1]. Humanity is now in desperate need of new, efficient antibiotics able to break the resistance of pathogenic bacteria species, preferably with new killing mechanisms to overcome existing bacterial defence systems. Unfortunately, the great boom in the early development of antibiotic chemistry was followed by a strong decline in the number of new antibiotics in the pipeline of medical companies [2]. Scientists are currently looking for new antimicrobial agents, and the hope may again come from nature.

AMPs, also known as host defence peptides, are short proteins (5–100 amino acids) found in a wide variety of life forms. AMPs are structurally diverse; they are positively charged proteins found in living organisms such as mammals, birds, insects, crustaceans, fish, plants and microbes [3,4]. AMPs were discovered in 1939 when Rene Dubos isolated an antimicrobial agent named gramicidin from a soil *Bacillus* strain, which protected mice from pneumococcal infection [5]. Afterwards, several AMPs were discovered in both prokaryotic and eukaryotic kingdoms [6]. To date, more than 3000 AMPs have been officially classified and registered in the AMP database [7]. It is worth noting that more than 300 AMPs were found in frog skin [6]. Natural AMPs have potent and broad-spectrum activity against Gram-positive and Gram-negative bacteria, protozoa, viruses (e.g., HIV, HCV), fungi and parasites [7,8,9,10,11,12,13,14,15,16], displaying bacteriostatic, microbicidal and cytolytic properties [17].

The scientific society is aware that we are imminently entering the post-antibiotic era. The previous worldwide pandemic events exposed the need for alternative antiviral curations. We were also witnesses to the frightening spread of the infectious “superbug fungus” specimen Candida auris with its mortality reaching around 40% [18].

The evolution of pathogen resistance to antimicrobials brings the applicability of this group of drugs under threat. The mean time usefulness of antibiotics has decreased a few times since 1995. Two complementary approaches have been proposed to fight or at least slow down the resistance evolution. The first is aimed at increasing the amount of new drug development parallelly to decreasing the time needed for drug approval by the authorities. The second mechanism relies on reducing antimicrobial misuse (i.e., new diagnostic approaches for more accurate drug prescriptions, and procedures for reducing pathogen transmission) [19]. As the World Health Organization (WHO) has announced, the alarming global rise in resistance to conventional antimicrobials represents a potential and serious risk to public health [20,21]. Moreover, the interest in AMPs has recently increased during the SARS-CoV-2 syndrome pandemic in the search for new antiviral molecules to counteract the COVID-19 disease [21,22].

Current scientific reports confirm the particularly important therapeutic function of the group of peptides belonging to AMPs and their anticancer properties. Cancer diseases cause high morbidity and mortality in people all over the world. According to the latest World Health Organization (WHO) reports, the number of people diagnosed with cancer diseases has doubled in the last decade. In the next two decades, the number of patients is expected to continue to increase at a similar rate. Therefore, the world of science is intensifying research on developing new biologically active molecules, which, combined with conventional therapies, would improve the effectiveness of treating cancer patients. Due to their unique features (anticancer activity), selected AMPs are able to selectively affect cancer cells, which makes them an important social research object.

The unique properties of some AMPs, such as their broad spectrum of activity, generally low toxicity to host cells, as well as reduced induction of resistance in target cells [23], are an excellent tool for the development of a new class of antibacterial, antiviral, antifungal and even anticancer agents. On the other hand, the growing interest in well-being, health and physical appearance has increased the demand for discoveries in the field of cosmeceutical products, i.e., those that combine cosmetics with science. In dermatology and cosmetology, substances with the properties of AMPs are constantly sought after.

Pharmaceutical companies are making efforts to commercialise AMPs in an attempt to capitalise on the potential that this class of drugs can represent in the $80 billion global anti-infective market [24,25]. Currently, several AMPs are in various stages of clinical trials, and several have been on the market for some time. The number of clinical trials related to AMPs is very optimistic when it comes to introducing new drugs to global markets [26]. However, there is also a reason for joy for the cosmetic market in connection with the development of these molecules for possible use in cosmetology. AMPs have been revealed to possess antioxidant, self-renewal and pro-collagen effects, which are desirable in the cosmeceutical field [27].

## 2. Classification

AMPs can be classified in many different ways, one of them being structural flexibility. There are four categories based on their secondary structure: linear α-helical peptides, β-sheet peptides with the presence of two or more disulphide bonds, β-hairpin or loop peptides with the presence of a single disulphide bond and/or cyclisation of the peptide chain, and finally, extended structures [6] (Table 1). In the available AMP databases, approximately 14% of AMPs are helical peptides, 4% are β-type peptides, and 4% are mixed-type peptides [28].

AMPs can also be classified by their source of origin: bacteria (200), archaea (4), fungi (13), plants (343), and animals (2159) [43]. Currently, the available publications indicate that scientists are mainly focused on AMPs isolated from the plant phylum. They have been isolated from the roots, seeds, flowers, stems and leaves of a wide variety of species. Plant AMPs are grouped into several families and share general features: thionins, defensins, lipid transfer proteins, hevein-like proteins, and cyclotides [44].

Thionins were first isolated in 1942 from fractions of wheat and barley as a group of low-molecular-weight amphipathic vegetable proteins [45,46]. Thionins consist of 45–48 amino acid residues (~5 kDa), contain 6 or 8 cysteines and 3 or 4 disulphide bonds, and are rich in arginine and lysine. Their structure includes two antiparallel α-helices and an antiparallel double-stranded β-sheet. These are positively charged peptides at neutral pH [47]. They form a ring structure topology because of the end-to-end disulphide bond that connects the N and C ends; therefore, they can be classified as cyclic peptides [48]. However, they are not true cyclic peptides because disulphide-bonded cysteines are not located directly at the N- and C-termini. Other cysteines in polypeptides can also form disulphide bonds [49,50,51]. Thionins have been identified in many plant species [46] and are toxic to bacteria, fungi and yeasts [52]. For example, species sensitive to purothionin isolated from wheat endosperm crude are bacteria such as *Pseudomonas solanacearum*, *Xanthomonas phaseoli*, *Xanthomonas campestris*, *Erwinia amylovora*, *Corynebacterium fascians*, *C. flaccumfaciens*, *C. michiganese*, *C. poinsettiae*, *C. sepedonicum* [53]. The species sensitive to α-purothionins isolated from wheat endosporum is the fungus *Rhizoctonia solani* [54]. Viscotoxin A3 and B isolated from the leaves and stems of *Viscum album L*. are toxic to the fungi *Fusarium solani*, *Sclerotinia sclerotiorum*, *Phytophtora infestans.* The study reported that a concentration between 3 to 10 µm induced membrane disruption, H_2_O_2_ production and spore death [55]. Thioin isolated from *Nicotiana attenuata* PR-13 is toxic to *Pseudomonas syringae pv* bacteria, which, according to the 2012 report of the journal Molecular Plant Pathology, ranks first on the list of the most dangerous bacterial plant pathogens [56].

Defensins are the best known and probably the largest family of all membrane-soluble plant AMPs [57]. The first plant defensins were isolated from wheat *T. aestivum* and barley *Hordeum vulgare* and contained four disulphide bonds, similar to α-thionins, β-thionins and γ-thionins. Defensins are positively charged, basic, cysteine-rich peptides with a mass of about 5 kDa [58]. However, γ-thionins were found to be structurally different from α-/β-thionins and were classified as plant defensins based on their similarities in sequence, structure and function to mammalian and insect defensins [59,60,61]. Defensins have a variety of biological functions, such as inhibiting microbial growth, inhibiting α-amylase and trypsin activity, influencing self-compatibility, mediating abiotic stress, and acting as epigenetic factors [62,63,64,65,66,67]. Plant defensins are best known for their antimicrobial properties against a wide spectrum of plant pathogens such as bacteria, yeasts, oomycetes and necrotrophic pathogens [68,69,70], anticancer and antiviral properties [71,72], and they interact with glucosylceramides in yeast and fungal membranes to induce cell death [49,50]. For example, in the in vitro study from 2007, human alpha-defensin 1 (HNP1) at the concentrations 1.25, 5, 10 and 20 μg/mL reduced viral haemorrhagic septicaemia virus infectivity to 86, 80, 57 and 19%, respectively [73]. NaD1, an ornamental tobacco defensin, demonstrated PIP2-mediated anticancer activity at 10 μM in the in vitro studies on the range of mammalian tumour cell lines [74]. Some human β-defensin 3 derivatives (hBD3), such as hBD3_A_ (full-length hBD3 with disulphide pairings Cys^11^-Cys^41^, Cys^18^-Cys^40^, and Cys^23^-Cys^33^), have antimicrobial activity similar to native hBD3 against *E. coli*, *S. aureus*, and *C. albicans* with LC90 values of approximately 5 μg/mL, 12 μg/mL, and 15 μg/mL [75].

Lipid transfer proteins (LTPs) are cationic peptides with a molecular weight of 7 to about 10 kDa. LTPs are classified as LTP1 (9 kDa) or LTP2 (7 kDa), depending on their molecular weight. They have eight Cys residues and have a low overall amino acid sequence similarity (about 30%). Almost all LTPs lack tryptophan residues, with the exception of a few isoforms in Arabidopsis and rice, which have 1–2 Trp [76]. LTPs can bind to a variety of lipids, including fatty acids, phospholipids, prostaglandin B2, haemolytic derivatives and acyl coenzyme A [77,78,79]. These peptides have been shown to reversibly bind and transport hydrophobic molecules in an in vitro model [77]. LTPs not only inhibit the growth of fungi and bacteria but also participate in plant defence systems [80,81,82,83].

For example, organisms sensitive to LTPs isolated from wheat (Sumai3) are the fungi *Rhizoctonia solani*, *Curvularia lunata*, *Alternaria* sp., *Bipolaris oryzae*, *Cylindrocladium scoparium*, *Botritis cinerea*, and *Sarocladium oryzae*, with the tested concentrations varying from 1.25 μg/μL to 3.75 μg/μL [84]. LTP-s1 and LTP-s2 isolated from spinach leaves are toxic to *Clavibacter michiganensis* subsp. *Sepedonicus* and *Pseudomonas solanacearum*. These proteins were found to exhibit an EC-50 in the range of 0.1–1 μM for the bacterial pathogens *C. michiganensis* and *P. solanacearum* [85]. AceAMP1 LTPs isolated from onion seeds have both antifungal and antibacterial properties [86].

Hevein-like AMPs are alkaline peptides that were identified for the first time in the latex of the *Hevea brasiliensis* rubber tree, presenting strong antifungal activity in vitro. They are peptides with a molecular weight of approximately 4.7 kDa [87,88]. These peptides contain a conserved chitin-binding domain with the amino acid sequence SXFGY/SXYGY, where X can be any amino acid residue [89,90,91]. The hevein domain consists of an antiparallel β-sheet and a short α-helix, and the scaffold is stabilised by 3–5 disulphide bonds [92]. Peptides from this group also have antibacterial activity and have been identified in various plant species [92,93,94,95]. The species susceptible to WjAMP1 isolated from leaves of *Wasabia japonica* are the fungi (IC_50_ between 5.8 to 80 μg/mL) *Botrytis cinerea*, *Fusarium solani*, *Magnaporthe grisea*, and *Alternaria alternata,* and the bacteria *Escherichia coli*, *Agrobacterium tumefaciens*, *Pseudomonascichorii*, *P. plantarii (Burkholderia plantarii)*, and *P. glumae (B. glumae)* [94].

Among all the AMPs, cyclic peptides (cyclotides) exhibit the greatest stability and resistance to proteolytic cleavage. Although peptides and proteins are biomolecules that have been investigated for decades, cyclic peptides have gained popularity only in recent years. Microbial peptides with a cyclised backbone were initially reported by Saether et al. in 1995 [96], although they had been used in traditional medicine long before that in Africa to accelerate labour and childbirth [44,96]. Cyclotides are a group of naturally occurring circular proteins that have been discovered in bacteria, plants and animals [97,98]. These are cyclic peptides containing 28–37 amino acids [99,100]. Cyclotides possess a cyclic backbone consisting of six loops, which are formed by six conserved cysteine residues arranged in a cross-linked and bonded manner [101]. This cysteine linkage is formed when the first two disulphide bonds (Cys1-Cys4 and Cys2-Cys5) and their fused backbone form a ring that is penetrated by the third disulphide bond, Cys3-Cys6 [102]. Among all the AMPs, cyclic peptides exhibit the greatest stability and resistance to proteolytic cleavage, which makes them potential therapeutic agents [103], including anticancer [104], anti-HIV [105], insecticide [106,107] and antimicrobial agents [101]. The cyclotide Vitri isolated from *Viola tricolor* demonstrated cytotoxicity to human lymphoma and myeloma cells. Cycloviolacin H4 isolated from *Viola hederaceae* was able to cause haemolysis in human erythrocytes [97]. Cycloviolacin O13, O14 and O24 isolated from *Viola odorata* showed anti-HIV properties at the concentrations (EC_50_) of 320, 440 and 308 nM (with cytotoxic concentration IC_50_ > 6.4, 4.8, and 6.2 μM, respectively) [108]. Kalata B1 from *Oldenlandia affinis* demonstrated insecticidal, molluscicidal, haemolytic, nematocidal, anti-HIV and antibacterial activity. Kalata B2 from *Oldenlandia affinis* demonstrated insecticidal, molluscicidal, nematocidal and antibacterial activity. The incorporation of these cyclotides into the *Helicoverpa punctigera* larvae diet caused ~50% mortality and a reduction in the size and growth rate of the survivors [107,108,109,110,111,112]. Haemolytic, antibacterial and anti-HIV properties were also exhibited by circulins A and B isolated from *Chassalia parviflora*. The cytoprotective concentrations of circulins against HIV viruses (EC_50_′s) ranged from 40 to 260 nM [113], whereas the cyclopeptide MCoTI-II from *Mormodica cochinensis* is a trypsin inhibitor (IC_50_ of 2.12 µM) [114,115]. Although peptides and proteins are biomolecules that have been investigated for decades, cyclic peptides have gained popularity only in recent years.

Different AMPs have been identified in avocado fruit and the fruits of Capsicium, which could be used in the treatment of infections caused by *Streptococcus aureus* and *Escherichia coli* strains [116,117,118]. Example structures of AMPs are shown in Figure 1.

## 3. Mechanism of Antimicrobial Peptide Action

Peptides can be categorised in multiple ways based on the type of activity (e.g., anticancer, killing bacteria, induction of angiogenesis, modulation of gene expression), mechanism of action, or structure and sequence. In this chapter, various mechanisms of action will be described and used to group the proteins.

To utilise AMPs in practical applications, it is important to understand their mechanism of action. There have been multiple research studies to understand how they interact with and impact other organisms. Initial studies showed that the proteins use membrane targeting. This is a different mechanism than that used by antibiotics [26,39,119,120], and the cell membrane itself also impacts the interaction with the protein as it depends on their lipids. Later research studies showed [39] there are also different mechanisms of action. Besides membrane targeting, there is also non-membrane targeting and immune modulation. The mechanisms of action are split into two groups. The first one, named direct killing, includes membrane targeting and non-membrane targeting (Figure 2 and Figure 3); the second is immune modulation (Figure 4).

### 3.1. Direct Killing

Individual AMPs interact with bacterial cell membranes, impacting the construction of its outer or inner membrane, leading to cell death. This is realised by electrostatic forces between the negatively charged bacterial cell surface [121,122,123] and the positively charged AMPs; therefore, this interaction depends on cell surface lipids that are negatively charged. AMPs accumulate at the surface and self-assemble on the bacterial membrane after reaching a certain concentration. There are four main types of AMP interactions with the cell membrane, named the barrel-stave model, toroidal-pore model, carpet model and aggregate model, graphically presented in Figure 3.

Bacteria are classified as either Gram-positive or Gram-negative, characterised by significant differences in their cell envelopes. The inner or cytoplasmic membranes of both bacteria groups are similar, but the outer cell envelopes are significantly different. In Gram-positive bacteria, there is a layer of cross-linked peptidoglycan decorated with negatively charged teichoic acid surrounding the cytoplasmic membrane, forming a thick matrix that maintains the rigidity of the bacterial cell. Nano-sized pores penetrate into the peptidoglycan layers, allowing AMPs to diffuse through [124]. In contrast, the peptidoglycan layer in Gram-negative bacteria is much thinner and less cross-linked. In addition, Gram-negative bacteria have an additional outer membrane outside the peptidoglycan layer. The inner layer consists purely of phosphate lipids, while the outer leaflet is primarily a coat of lipopolysaccharides [125]. LPS molecules are decorated with a high number of negatively charged phosphate groups that are engaged in salt bridges with divalent cations (e.g., Ca^2+^ and Mg^2+^), resulting in an electrostatic network [126]. This electrostatic region serves as a primary barrier to most hydrophobic antibiotics, resulting in low permeability. Therefore, the details of how AMPs penetrate into Gram-positive and Gram-negative bacteria must vary in their atomistic interactions [127]. In the case of Gram-positive bacteria, AMPs need to diffuse across the peptidoglycan matrix first and then act on the cytoplasmic membrane. In contrast, killing Gram-negative bacteria involves the perturbation or disruption of both the outer and cytoplasmic membranes. The inability to permeabilise or disrupt the outer membrane results in the loss of antimicrobial activity. Daptomycin is able to disrupt the cytoplasmic membrane but not able to permeabilise/disrupt the outer membrane of Gram-negative bacteria. As such, it is highly active against Gram-positive bacteria such as methicillin-resistant *Staphylococcus aureus* (MRSA) but has no activity against Gram-negative bacteria [128].

Another way of impacting cells is non-membrane targeting. In this case, AMPs penetrate the cell and interact with the cell interior by inhibiting DNA, RNA and protein synthesis, impacting protein folding, enzyme activity and cell wall synthesis, leading to cell death.

### 3.2. Immune Modulation

AMPs can recruit and activate immune cells (Figure 4). This results in the control of inflammation and increased cell killing [129,130,131]. AMPs can also produce a variety of immune responses: the activation, attraction, and differentiation of white blood cells; the stimulation of angiogenesis; the reduction of inflammation by lowering the expression of proinflammatory chemokines; and controlling the expression of chemokines and reactive oxygen/nitrogen species [129,130,131,132,133,134]. Moreover, many immune cells also produce AMPs (e.g., neutrophils and macrophages), and they can be the first line of defence against invading microbes [135].

### 3.3. Mechanism of Action against Cancer Cells

AMPs, due to their cationic properties, bind preferentially to cancer cells, causing the disruption of their lipid membranes, thus leading to cell death (apoptosis) [136]. AMPs bind more strongly to the membrane of bacterial cells than to the membrane of normal eukaryotic cells; however, the surface of cancer cells differs in some features from normal eukaryotic cells. Cancer cells have much more negatively charged particles on their surface than normal cells. In addition, due to the large number of villi, these cells have a larger surface area, which allows more molecules of antibacterial peptides to bind to the membrane. Differences in the structure of cell membranes, as well as the large surface of cancer cells with villi present on the surface, may cause cancer cells to be penetrated with greater selectivity by AMPs than normal cells [137,138,139]. Some AMPs can induce apoptosis, or programmed cell death, in cancer cells by activating various pathways involved in cell death. This can lead to the elimination of cancer cells without damaging normal cells. Angiogenesis is the process by which new blood vessels are formed, and it is essential for the growth and spread of cancer cells. Some AMPs can inhibit angiogenesis by interfering with the signalling pathways involved in blood vessel formation, thereby limiting the growth and spread of cancer cells. Other AMPs can modulate the immune response by activating immune cells such as natural killer cells and macrophages, which can help to eliminate cancer cells. AMPs can also stimulate the production of cytokines, which are important for the regulation of the immune response [140]. Figure 5 presents the different mechanisms of AMP anticancer action.

## 4. Possible Applications in Pharmaceutical, Biomedical and Cosmeceutical Fields

AMPs with broad-spectrum antibacterial, antiviral, antifungal and anticancer activity are expected to become alternative antibiotics through the development of AMP-based therapies. Currently, several AMPs have been approved for antibacterial treatment by the Food and Drug Administration (FDA), and other AMPs are under clinical development [141]. Despite the promising potential of AMPs as medical therapeutics, there are many challenges that need to be overcome. The limitations to the intravenous administration of AMPs are caused by enzymatic degradation in blood plasma due to a short half-life. Oral application is also limited due to the pre-systemic enzymatic degradation of the peptides and poor penetration into the intestinal mucosa. In clinical trials, AMPs are mainly limited to topical applications due to their different enzymatic degradation, systemic toxicity and rapid hepatic and renal clearance [142,143]. Consequently, the local application of AMPs is the most common administration route, including delivery via topical dermal creams and skin softeners. In order to improve the AMP delivery system, polymeric materials such as hydrogels [144], chitosan [145], and hyaluronic acid [146] are used. AMPs may be covalently linked or non-covalently encapsulated in delivery systems. The covalent attachment of polyethylene glycol in a biomolecule (PEGylation) can reduce non-specific tissue uptake, cellular toxicity, and increase blood half-life and proteolytic degradation [147,148]. The conjugation of AMPs to hyperbranched polyglycerol (HPG) provides a better antimicrobial effect [149]. Lipids and surfactants can also be used as a conjugate to protect the peptides under extreme alkaline/acidic conditions and elevated temperatures [150]. Mesoporous silica particles [151], quantum dots [152], gold and silver nanoparticles [153,154], titanium [155], graphene [156] and carbon nanotubes [157] have also been used.

The high cost of peptide production also limits the commercial and clinical development of AMPs. The cost can be reduced by obtaining recombinant AMPs in prokaryotic or yeast expression systems with the usage of genetic engineering methods.

### 4.1. Pharmaceutical Applications

AMPs have several potential pharmaceutical applications due to their ability to kill or inhibit the growth of pathogens. Some AMPs have been found to be effective against Gram-negative and Gram-positive bacteria, including multidrug-resistant strains. They can also prevent the formation of bacterial biofilms, which are a common cause of chronic infections. Some AMPs have shown promise in treating fungal infections, with special emphasis on immunocompromised patients, such as those with HIV/AIDS. AMPs may work as antiviral drugs by disrupting viral entry, inhibiting viral replication, or promoting immune responses. The studies show that AMPs can work profitably with chemotherapy drugs. Some of them exhibit anticancer properties by inhibiting the growth and proliferation of cancer cells. AMPs have been shown to promote wound healing by accelerating the formation of new blood vessels and the migration of skin cells to the wound site. Moreover, they can be used as vaccine adjuvants and as natural preservatives [158].

#### 4.1.1. Antibacterial Activity

The skin infection segment held the largest revenue share of 30.3% in 2021 and is likely to dominate the market during the forecast period. The growth of the skin infection segment is augmented by the rising prevalence of skin infections such as cellulitis, impetigo, furuncles, carbuncles, and others, and the wide availability of products for the treatment of bacterial skin infections. For instance, CUBICIN RF (daptomycin for injection), a type of lipopeptide product manufactured by Merck & Co., Inc., is used for the treatment of paediatric patients and adults with complicated skin and skin structure infections caused by *Streptococcus aureus*, *Streptococcus pyogenes*, and *Streptococcus agalactia* [159]. Daptomycin’s t½ is relatively long (~9 h), which allows once daily dosing in patients. This compound’s maximum dose ranges from 6 mg/kg up to 8 mg/kg, in which the linear pharmacokinetics is maintained up to 6 mg/kg.

However, the bloodstream infection segment is anticipated to register the fastest growth rate during the forecast period. The growth of this segment is augmented by the rising incidences of bloodstream infections, rising awareness about bloodstream infections, and the availability of a robust product portfolio for the management of bloodstream infections. For instance, Polymyxin B vials containing 500,000 units/vial manufactured by Xellia PHARMACEUTICALS are used for the treatment of bloodstream infections due to strains such as *E. coli*, *Pseudomonas aeruginosa*, and *H. influenza* [160,161].

Vancomycin and bacitracin are both antibiotics that are common AMPs from natural sources interfering with Gram-positive bacteria cell wall synthesis [162,163]. Vancomycin has been in clinical use since the 1950s and is approved for use in many countries worldwide under various brand names, including Vancocin, Vancomycin Hydrochloride, and others. It can be administered as an intravenous injection (15 to 20 mg/kg every 8 to 12 h), a capsule (125 to 500 mg four times daily), or in the case of skin infections, as a topical formulation (concentrations in the range of 1–2%) [164,165,166]. Bacitracin vials manufactured by Xellia Pharmaceuticals are indicated for the treatment of infants with pneumonia and empyema (50,000 units or 100,000 per ml of solution per vial). Bacitracin is an example of a polypeptide product that works by inhibiting bacterial cell wall synthesis. It is often used topically in ointments and creams [161,167,168]. Dalbavancin, primarily sold by Melinta Therapeutics, received FDA approval in May 2014. Since then, the drug has been marketed in the US and Europe. Sold under the brand name Dalvance/Allergan (500 mg/vial), Dalbavacin is used for treating acute bacterial skin and skin structure infections (dosage between 18 and 22.5 mg/kg with maximum 1500 mg) [169,170,171]. In 2018, Melinta reported global net product sales of $45,9 million for Dalvance [172]. It appears to be very effective in many serious Gram-positive infections. A long half-life and good diffusion in bone tissue suggest that Dalbavacin could be effective in the treatment of prosthetic joint infections. Even though AMPs are considered a cure for antibiotic-resistant bacteria, such as *Staphylococcus aureus*, some enterococci can gain resistance to this peptide, which is particularly concerning as they can cause serious infections that are difficult to treat [173]. Fortunately, some semi-synthetic peptide derivatives can be a solution in this case. Telavancin, sold by Theravance Biopharma, apart from skin infections, is also approved for the treatment of hospital-acquired and ventilator-associated bacterial pneumonia (HABP/VABP; brand name Vibativ) [174]. Vibativ is indicated for skin infections and pneumonia, which share the same dosage of 10 mg/kg [175]. Similar to Vancomycin, Dalbavancin and Telavancin work by inhibiting bacterial cell wall synthesis. Oritavancin, another example of a semi-synthetic lipoglycopeptide, shares similar properties, but has a unique action mechanism. It can bind and disrupt bacterial cell membrane integrity, which is a beneficial antimicrobial property. It is FDA-approved for acute bacterial skin and skin structure infections under the brand name Orbactiv [176].

In September 2021, AuroMedics Pharma LLC announced that it received approval from USFDA to manufacture Daptomycin for an injection to treat serious bacterial infections, including skin and soft tissue infections, bloodstream infections, and endocarditis. Daptomycin is a lipopeptide antibiotic under the brand name Cubicin, administered intravenously (4 mg/kg) [177]. It works by disrupting the bacterial cell membrane. The same mechanism is shared by gramicidin, which is usually formulated in combination with other antibiotics. It is an antibiotic agent for the topical treatment of skin infections, with the drawback that the oral administration of this drug can be toxic. In vitro and in vivo studies have shown the great potential of this drug as a therapeutic agent in renal cell carcinoma, the most common type of kidney cancer in adults [178].

AMPs are reserved for use in situations where other antibiotics may not be effective. However, the use of antimicrobial peptides should be guided by susceptibility testing and other factors, similar to antibiotics, to minimise the risk the bacteria gaining resistance [21].

#### 4.1.2. Antiviral Activity

Some AMPs have been shown to have antiviral activity against various viruses, including both enveloped and non-enveloped viruses [179]. There are nine peptidomimetic drugs on the market for the treatment of AIDS and at least four in clinical development for the treatment of HCV infections. Saquinavir, a peptidomimetic protease inhibitor, is a molecule with a hydroxy ethylene scaffold that mimics the typical peptide bond but is not broken down by HIV-1 protease. HIV (human immunodeficiency virus) is a target for antiviral peptides (AVPs) as they can be designed to target specific components of the virus, such as the fusion process and protease enzyme [179]. Enfuvirtide (T-20) is a synthetic peptide drug that is the first FDA-approved viral peptide inhibitor [180]. It is used for HIV infection and can be used to treat infections resistant to antiretroviral drugs HIV-1 derivative. The mechanism of action of Enfuvirtide is binding to glycoprotein 41 (gp41) in a way that prevents the conformational change required for fusion. As a result, this prevents the virus from entering the host cell and replicating. Enfuvirtide is administered by subcutaneous injections, which can lead to the frequent occurrence of painful injection site reactions. Hepatitis C is another viral infection that can be treated with antiviral peptides. Currently, two drugs are approved for that disease: a semi-synthetic peptide telaprevir and a synthetic peptide boceprevir [181]. They belong to a class of medications called protease inhibitors and have the same target. They block the replication of some types of hepatitis C viruses by targeting the NS3/4A protease enzyme, leading to the reduction of the virus in the blood, which results in improved liver function. These drugs are typically used in combination with PEGylated interferon and ribavirin. Since the COVID-19 outbreak, numerous antiviral peptides and peptidomimetics against SARS-CoV-2 have been reported. Although no peptide antiviral drugs for COVID-19 have entered clinical trials, some FDA-approved peptide drugs have been recommended for clinical trials for COVID-19 through virtual screenings and in silico drug repurposing methods. Researchers have shown that Enfuvirtide could inhibit SARS-CoV-2 entry into host cells with great potency and recommended it for COVID-19 clinical trials. Peptide-like small molecules, amino-acid-like derivatives, and peptidomimetics such as remdesivir and lopinavir have been utilised in COVID-19 clinical trials and treatment [182].

#### 4.1.3. Anticancer Activity

Due to the resistance of cancer cells to treatment and the toxicity of cytostatics, new possibilities for anticancer therapies are constantly being sought. This has led the focus on AMPs, which have the ability to resist cancer growth. Zhao reported the anticancer activity of the HPRP-A1 peptide isolated from *Helicobacter pylori* [183]. Further, the combined effect of iRGD (homing peptide) and HPRP-A1 were examined for their enhancement of anticancer activity. Furthermore, the results suggested that iRGD helped to improve the penetration of HPRP-A1 into A549 MCS [184]. L-K6 was reported to be capable of killing MCF-7 breast cancer cells via nuclear disruption without cell surface disruption [185]. AMPs with the ability to inhibit cancer cell growth are mentioned in Table 2.

#### 4.1.4. Broad Spectrum of Antimicrobial Activities

Despite promising results from preclinical and clinical studies, some AMPs require further work before being approved for pharmaceutical use. LL-37 is a naturally occurring AMP produced by immune cells, such as neutrophils and epithelial cells. Although it is not an FDA-approved drug, it has broad-spectrum activity, including in infectious diseases, inflammatory disorders, and cancer. LL-37 has several undesirable properties, such as possible bacterial resistance, cytotoxicity, and the inability to retain antimicrobial activity in the environment [196]. Clinical studies showed that LL-37 could be a potentially effective treatment option for patients with large ulcers [197]. Defensins are a family of small, cationic peptides found in various tissues and organs of the human body. They play a key role in the innate immune system by protecting against invading pathogens. They work by disrupting cell membranes, leading to cell lysis. In the review from 2019 on diabetic foot ulcers, it was suggested that beta-defensin-2 (hBD2) could be assessed as a drug for that disease [198]. Omiganan pentahydrochloride, a synthetic analogue of human defensin, was clinically tested for the treatment of atopic dermatitis, with significant results in reduction of the Scoring Atopic Dermatitis (SCORAD) index, showing it can be a safe and effective treatment option [199,200]. Nisin is a peptide produced by the bacteria *Lactococcus lactis.* Commonly used as a food preservative, it is now in clinical studies for bacterial infections. It works by membrane depolarisation, leading to the leakage of cellular contents and ultimately cell death. Clinical studies showed that nisin can be effective in the treatment of various bacterial infections, including diabetic foot ulcers and *Helicobacter pylori* infection [201,202,203]. Melittin is a naturally occurring peptide that is found in *Apis mellifera* venom. It has been studied for its therapeutic uses, including as an antimicrobial agent and as a potential cancer therapy [204,205]. Histatin-1 is a cationic peptide found in human saliva. It works by disrupting the cell membrane of microorganisms, causing cell lysis. It has been shown to be effective against a variety of oral pathogens, including *Candida albicans*, *Streptococcus mutans*, and *Porphyromonas gingivalis* [206,207]. Promising antimicrobial properties have been exhibited by pyocins derived from *Pseudomonas aeruginosa.* They have been successfully used in vivo in mice peritonitis treatment [21].

Obesity is a major health problem worldwide and is associated with numerous health risks, including type 2 diabetes. According to a report by Market Research Future, the obesity treatment market size is projected to grow over 31 Billion USD by 2030, showing a 16.70% compound annual growth rate (2023–2030) [208]. Inhibitors of pancreatic lipase activity are being investigated as potential treatments for obesity and related metabolic disorders. Several peptides isolated from soybean have demonstrated properties limiting the activity of that enzyme and are being considered for clinical trials [21].

Summing up, AMPs have various pharmaceutical applications, such as killing or inhibiting pathogen growth, preventing bacterial biofilms, treating fungal infections, working as antiviral drugs, and promoting wound healing. The market is dominated by the skin infection segment, while the bloodstream infection segment is projected to have the highest growth rate during the forecast period. Semi-synthetic peptide derivatives have been developed to address resistance issues; however, susceptibility testing should guide AMP use to minimise the risk of resistance.

### 4.2. Biomedical Applications

Modern healthcare uses various medical devices that improve or restore the function of the human body. This significantly improves the life quality of individuals affected by injuries or diseases [209] and generates demand for new technologies and special materials, which over the last decades have resulted in the development and popularisation of medical devices or biomaterials such as catheters [210], pacemakers [211], hip implants and prosthesis [212], and contact lenses [213]. All of these devices confer many benefits to patients but concurrently introduce the risk of microbial colonisation and infections due to foreign material being introduced to the body [214,215]. Figure 6 illustrates the biomedical applications of AMPs.

#### 4.2.1. Implantable Devices

Medical treatments utilising various implants can cause infection by introducing microorganisms to the human body that are attached to the implantable devices, or the patients can be infected during hospitalisation [216,217]. Antibiotics are broadly used to protect patients from infection consequences that can be very serious; however, their overuse and improper use have caused the growth of antibiotic resistance [218,219]. Additional protection is provided by various coatings, a thin layer of material on the selected surface intended to improve its properties or create a protective layer against harmful factors [220], such as a shield against bacteria [221], fouling [222], UV light [223], and corrosive substances [224].

To prevent implantable devices from becoming infected, antimicrobial-releasing coatings are preferred, as the agent also reaches the peri-implant tissue [225]. Hydrogel-based AMPs proved to exhibit strong antimicrobial activity against *Porphyromonas gingivalis*, a major cause of peri-implantitis, with no signs of toxicity [226]. Additionally, a gelatine-based hydrogel deposited on Ti surfaces, which allows the controlled release of the short cationic AMP HHC36, is another example. AMP release prevented *S. aureus*, *S. epidermidis*, *E. coli*, and *Pseudomonas aeruginosa* biofilm formation [227]. The AMP HHC-36 sustained-release PDLLA-PLGA coating on TiO_2_ nanotubes maintained an effective drug release for 15 days in vitro and showed significant antiproliferative activity against *Streptococcus aureus.* In addition, in vivo studies demonstrated that the coating was biocompatible and antibacterial [228]. In another similar approach, GL13K-eluting coatings on TiO_2_ nanotubes prevented the growth of *Fusobacterium nucleatum* and *Porphyromonas gingivalis* [229]. A PCL-based dual coating proved the sustained antibacterial functionality of HHC36 for 14 days. The coating was translated onto silicone urinary catheters and showed promising antibacterial effectiveness when compared with the commercial silver-based Dover catheter [230]. Another scientific group modified PLA films by gallium implantation and subsequently functionalised them with hBD-1. Ga and defensin independently and synergistically contributed to the creation of a novel antimicrobial surface, which significantly decreased the total live bacterial biomass [231]. Melittin was physically stabilised on chitosan, chitosan/Vancomycin and oxacillin antibiotic coatings applied to etched Ti implants. The antimicrobial characteristics of the coatings and the synergistic effect of Melittin and antibiotics against MRSA and Vancomycin-resistant S. aureus (VRSA) were evaluated in two states: floating and adherent to the implant’s surface [232]. For orthopaedic and dental applications, a bioactive coating (Pac@PLGA MS/HA coated Ti) was deposited on the Ti surface Pac@PLGA MS/HA-coated Ti exhibited a cytotoxic effect on *E. coli* and *S. aureus* [233].

#### 4.2.2. Biomedical Devices

The coatings can also enhance medical devices with multiple biofunctions such as drug delivery [234], biosensing [235], antibacterial properties [236], and osseointegration [237]. Devices with coatings can much better fulfil surgical and clinical requirements; therefore, the pharmaceutical and biomedical industries are constantly looking for advanced coatings with different functionalities. Many types of non-adhesive and antimicrobial coatings based on AMPs have been researched and tested [238,239]. These coatings can be divided into three groups: antifouling, contact-killing, and incorporating and releasing antimicrobials [240,241]. Chemical techniques are commonly used in contact-killing surfaces to immobilise AMPs to prevent microbial colonisation [242]. The structural properties of the peptides that are important for their antimicrobial activity should not be changed by the immobilisation process. Important parameters for AMP immobilisation include the orientation of the immobilised peptides and the AMP surface density, and the extent, flexibility, and spacer type for making the peptide–surface connection [243]. An example of a contact-killing surface is the hydrogel network with the covalently attached stabilised inverso-CysHHC10 peptide [244]. This coating exhibits antimicrobial activity in vitro against *Streptococcus aureus*, *Streptococcus epidermidis*, and *E. coli*. Additionally, brush-coating molecules may contain functional groups with antimicrobial activity, for example, through conjugation with the Tet20 [245] and Tet213 [246] AMPs. Another example is a polyurethane (PU) with a brush coating tethered to E6 AMP to avoid catheter-associated infection [247]. Chimeric peptide-modified Ti surfaces significantly reduced the adhesion of *Streptococcus aureus*, *Streptococcus epidermidis*, *P. aeruginosa*, and *E. coli* strains compared to bare Ti. Dental implants with immobilised GL13K on the Ti surface enabled osseointegration [248].

#### 4.2.3. Multifunctional Coatings

The creation of multifunctional coatings by combining the arginylglycylaspartic acid (RGD) cell adhesion sequence with lactoferrin-derived LF1-11 resulted in cell integration in vitro and the inhibition of *Streptococcus sanguinis* colonisation [249]. Another study reported a self-assembling coating of recombinant spider silk protein combined with Magainin I, which had the effect of reducing the number of viable bacteria on the coated surfaces [250]. Furthermore, Magainin II, which was covalently bonded to stainless steel surfaces, showed antibacterial activity against strains of *Streptococcus aureus* and *E. coli*. The surface modified in this way reduced biofilm formation and the amount of bacteria on the stainless steel surface [251]. An example of antifouling surfaces is Tet20 and E6 being coupled to poly(DMA-co-APMA) copolymer brushes attached to polystyrene nanoparticles (NPs) by Yu et al. [246,252]. These AMP-functionalised coatings acted against *P. aeruginosa* and *S. aureus*, but the coatings were less operative than the sole AMPs in solution. Furthermore, Muszanska et al. created polymeric brushes by dip-coating AMP-functionalised block copolymer Pluronic F-127 onto a silicone rubber surface. The surfaces prevented *Streptococcus aureus*, *Streptococcus epidermidis*, and *Pneumonia aeruginosa* colonisation and killed surface-adhered bacteria [253].

Monteiro et al. conjugated the peptide Chain201D and EG4-SAM control peptide to carbonylimidazole-activated tetra(ethylene) glycol-terminated self-assembled monolayers (EG4-SAM) onto gold surfaces. Compared to the control peptide, Chain201D killed a high proportion of adherent *S. aureus* and *E. coli* [254]. Another interesting study is surface-functionalised PU (PU-DMH) comprising PDMAPS brushes as the lower layer and HHC36 peptide-conjugated poly(methacrylic acid) (PMAA) brushes as the upper layer. The PU-DMH surface exhibited bactericidal properties against *E. coli* and *S. aureus* bacteria, preventing the accumulation of bacterial debris on the surfaces. The functionalised surface possessed persistent antifouling and bactericidal activities, both under static and hydrodynamic conditions. The microbiological and histological results of animal experiments also verified its in vivo anti-infection performance [255].

Ti surfaces were coated by Acosta et al. with engineered protein (elastin-like recombinamers; ELR) containing D-GLI13K via silanisation [256]. The biofilm formation was reduced by 90% due to the presence of AMPs on the ELR coatings, and the viability of *Streptococcus gordonii* and *Porphymonas gingivalis* in the adherent population was significantly reduced. In a recent study, hydroxyapatite (HA) nanorods co-doped with Fe and Si were fabricated on a Ti surface. The AMP HHC-36 was chemically attached to nanorods with and without polymer brushes. The polymer-brush-grafted HHC-36 reduced >99.5% of *Streptococcus aureus* and *E. coli* bacterial strains. This activity was attributed to the collaborative effect of AMP and the physical puncturing by HA nanorods. The in vivo studies performed on HA nanorods with the polymer-brush-grafted HHC-36 showed a reduction in the inflammatory response and the inhibition of bacterial infection [230].

#### 4.2.4. Water Purification Membranes

The shortage of clean drinking water has been a serious problem worldwide in recent years; therefore, the emerging application of polymer–bioactive molecule complexes has become a “hot” topic. Bacteria are present in almost every environment, especially in water, and antifouling membranes and surfaces have been prepared [257]. In this respect, composite membranes were synthesised by impregnating Ag NPs in the *N*-alkylated ter-polymer of poly(acrylonitrile), poly(*n*-butyl acrylate), and poly((2-dimethyl aminoethyl) methacrylate)), followed by cross-linking by the reaction with hydrazine hydrate. The antimicrobial activity of the composite membranes and a pristine membrane was determined by disc diffusion experiments on *E. coli* bacteria. The bacteria were drastically reduced (106 times) on the Ag-NP-containing membranes compared to the control [258].

#### 4.2.5. Detection Biosensors

Detection biosensors are another area of AMP biomedical applications. Thionins are also used to develop biosensors for diabetics that detect glucose levels. Salimi and co-workers confirmed that thionin induced in multi-walled carbon nanotubes selectively detects glucose [259]. Another example is a lung cancer biomarker probe with a low detection limit. The biosensor was created using a graphene oxide–thionin–hemin–Au nanohybrid. In this case, graphene oxide acted as a supporting material in which thionin and hemin were immobilised, followed by a reduction of gold particles by thionin [260].

### 4.3. Cosmeceutical Applications

The increased worldwide demand for improving physical appearance, health and well-being is driving research studies intended to develop new cosmetics. The most commonly needed substances are for anti-ageing purposes such as the prevention or reduction of wrinkles and skin smoothing, but also improving skin tone or reducing whelk effects. To obtain such effects, modern cosmetics have to be capable of blocking ion channels, providing antioxidants and anti-inflammatory effects, reducing melanin synthesis or tyrosinase inhibition, inducing cell proliferation/renewal, and many others. According to the current knowledge, AMPs are able to provide such effects. Additionally, the topical application of AMPs is the most common administration route and is also mostly desired for cosmetic applications. This causes peptides to be attractive ingredients in science-based cosmetics and to play an important role in the cosmetic industry [261,262]. 

There are examples of promising trials of AMP applications in skin treatment. An AMP designated CopA3 was used to prevent the ultraviolet-induced inhibition of type I procollagen synthesis and inhibited the induction of matrix metalloproteinase-1 in human skin fibroblasts, showing the potential for antiwrinkle cosmetic ingredients [263]. Several studies showed the capacity of the peptide LL-37 to suppress excessive collagen synthesis, providing an antifibrogenic effect [264,265,266]. Other AMPs, such as IDR-1018 [267], SHARP1 [268], β-defensin-1, -2, -3 [269,270], human neutrophil peptide α-defensins (HNPs) [271], and DRGN-1 [272] have been successfully used for treating wounds and improving effective tissue regeneration without scarring. 

Some AMPs have anti-inflammatory properties and can modulate the expression of cytokines, chemokines, and leukocyte activation [273]. Other AMPs such as indolicidin and its analogues [274], MC1-1 [275], sibaCec [276], SET-M33 [277], temporin-1TI and its analogues [278], hBD-3 [279,280,281], and LL-37 [282,283] have been shown to inhibit inflammatory responses. The peptides A3-APO successfully decreased the bacteria burden and reduced inflammation in acne [284]. The peptides (D4k) ascaphin-8 and (T5k) temporin-DRa were also effective at inhibiting the growth of *Propionibacterium acnes*, a pathogen resistant to the current antibiotics, and can be used to treat acne vulgaris [285]. Another positive effect was observed in modulating cellular renewal. AMPs have been shown to promote keratinocyte proliferation, including LL-37, which also promotes cellular migration and regenerative potential [286,287]. AMPs showing similar effect are IDR-1018 [288], PR-39 [23], human α-defensins (HNPs-1, -2, and -3) [289], human β-defensins [290,291,292], DAL-PEG-KSLW, KSLW [293], epinecidin-1 [294], psoriasin (s100a7) and koebnerisin (S100A15) [295]. Angiogenesis is an important aspect of skin appearance and health. Disorders of angiogenesis may cause rosacea, redness and vascular insufficiency [296].

Peptides such as AG-30/5C [297], LL-37 [286,298], PR-39 [299]; CRAMP [300], IDR-1018 [301], as well as α- and β-defensins [286] have the potential to counteract the age-induced decrease in angiogenesis in the skin and other tissues [290]. Peptide LfB17-34 can be used for skin whitening as it strongly increases melanin synthesis, which is associated with the elevated expression of the melanogenic enzymes tyrosinase and Trp1 [302].

Contributing factors to the ageing of the skin are reactive oxygen species. The intracellular formation of free radicals is influenced by ultraviolet light, ionising radiation, pollutants and diet [303]. Some peptides isolated from different fishes and mollusc species act as potential antioxidants [263,304,305,306,307,308,309]. Antioxidant activity was confirmed for several AMPs: temporin-TP1, brevinin-1TP1, brevinin-1TP2, brevinin-1TP3, brevinin-1LF1, palustrin-2GN1 [310]. 

The latest research studies of Unilever corporation patented in 2022 resulted in a novel and innovative approach. The external application of hydroxy stearic acid induced the secretion of AMPs from keratinocytes in the human body that act against bacteria infecting the skin, such as *Streptococcus aureus* and *Pseudomonas aeruginosa* [311].

## 5. Methods of AMP Production

Currently, the most commonly used method of obtaining AMPs, apart from extraction, is chemical synthesis. In clinical trials and commercial markets, large quantities of AMPs are needed to fulfil basic scientific study requirements. Isolation from natural sources and chemical synthesis are not cost-effective. In addition, the synthesis of longer peptides with more than 50 amino acids is not favoured [312]. For the large-scale and cost-effective production of large peptides and proteins, biological systems such as bacteria and yeast are required [313,314,315,316,317,318,319,320]. Biological systems do not need expensive pharmaceutical ingredients or toxic chemical reagents and solvents. *Escherichia coli* and yeast are two major systems used to produce recombinant antimicrobial peptides, accounting for over 95% of all reported cases [321,322]. Many antimicrobial peptides have been produced in yeast with good yields, but several others have been expressed in negligible amounts or obtained in inactive forms [323,324]. A bacterial expression system is more often used for obtaining recombinant antimicrobial peptides than a yeast expression system [322].

However, there are still some issues remaining, such as the toxicity of the expressed AMPs for host cells, which can be overcome by using plants as expression platforms [325,326,327]. Transgenic plants can be used for the oral delivery of AMP-based therapeutics [328,329]. In 2019, Da Sol Kim et al. [39] introduced a new expression vector for AMP production, in which recombinant AMP can be obtained in bacteria and plants. The vector is designed to work in prokaryotic systems and can be used to transform chloroplasts for large-scale AMP production. A plant expression system may offer several other advantages over microbial expression systems, including no risk of endotoxin contamination and the oral delivery of bioencapsulated therapeutics using edible plants [328,329,330,331]. The number of transgene copies in chloroplasts can be multiplied up to 10,000 per single plant cell [332,333], which leads to high expression of the transgene. Based on their research, the authors described a new expression platform for the efficient production of AMPs, which has the potential to function in both bacteria and plant chloroplasts.

Summarising, over many years, scientists have developed methods of obtaining AMPs not only through their direct isolation from organisms, but also chemical methods for the synthesis of these peptides, and finally more efficient methods of obtaining recombinant AMPs by genetic engineering.

## 6. Market

The global antimicrobial peptide market size was valued at USD 5 billion in 2021 and is expected to reach USD 7.85 billion in 2029. Despite multiple challenges associated with practical application of AMPs, there are also multiple advantages. The rising frequency of infectious diseases and the rising demand for effective and safe medicines are driving the expansion of the antimicrobial peptide market. The potential application to the treatment of hepatitis *C*, pneumonia, bacterial infections, HIV, and cancer is likely to increase the public acceptability of the treatments. As for today, antimicrobial peptide treatments are a promising newcomer in the field of immune modulation and antifungal drugs, and are gaining popularity as therapeutic agents for a variety of ailments, including skin infections. Therefore, due to the increased awareness of medications and therapies incorporating AMPs, the global market for AMPs is expected to rise significantly.

AMPs are now undergoing clinical trials, but the stability and half-life of these peptides in vivo are still not well-identified. These issues are limiting the global growth of the market; however, with a growing pool of scientists and talented technocrats interested in the arena of innovative therapeutics containing antimicrobial peptides for rare diseases, institutional research is substantially invested.

The hospital pharmacy segment held the highest revenue share of 45.4% in 2021, whereas the retail pharmacy segment is expected to register the fastest growth rate. Rising prescriptions of peptide antibiotics owing to various kinds of bacterial skin infections and ophthalmic infections is a major factor expected to drive the retail pharmacy segment growth. However, the growth of the hospital pharmacy segment is augmented by the rising prevalence of bloodstream infections, hospital acquired infections, and prolonged hospital stays.

The market for peptide antibiotics is expected to witness growth opportunities owing to the rising incidence of central-line-associated bloodstream infections. According to the Australian Institute of Health and Welfare, during 2019–2020, 700 Australian public hospitals reported 1428 cases of *Staphylococcus aureus* bloodstream infections. Amongst them, most of the infections were methicillin-sensitive *Staphylococcus aureus* infections. All types of bloodstream infections are serious infections and can cause prolonged hospital stays. Thus, the surge in the prevalence of bloodstream infections is likely to support market growth.

The rising incidence of antimicrobial resistance is also expected to provide lucrative growth opportunities for the peptide antibiotic market. For instance, according to the Centers for Disease Control and Prevention (CDC) report on antibiotic resistance threats in the US in 2019, 2.8 million antimicrobial resistance cases are recorded in the United States every year with about 35,000 deaths every year. Moreover, the CDC is implementing funding activities through their Antimicrobial Resistance Initiative, which supports over 50 state health departments and various local health departments. The CDC has also collaborated with state and local health departments, federal agencies, and the private sector to mitigate the antimicrobial resistance threat.

From 2016 to 2020, the CDC has invested in over 330 novel antibiotic resistance projects across 30 countries to combat antimicrobial resistance. Ongoing product approvals and strategic initiatives undertaken by key players such as partnerships, collaborations, product launches, and expansions are expected to cater to market growth.

Moreover, in July 2022, Boehringer Ingelheim, Evotec SE, and bioMérieux announced that they have created a joint venture for making the next generation of antimicrobials to fight against antimicrobial resistance. Furthermore, the availability of a robust product portfolio of peptide antibiotics is also anticipated to positively impact the growth of the market for peptide antibiotics [334,335,336].

## 7. Conclusions

Antimicrobial peptides have many promising properties for pharmaceutical, biomedical and cosmeceutical use. There have been multiple research studies to understand how they interact and impact other organisms that are required to utilise AMPs in practical applications.

The increasing antimicrobial resistance to antibiotics has created a need for new antimicrobial agents and AMPs to provide a potential solution due to their broad spectrum of activity against various microorganisms. AMPs can be used as new therapeutics, and there is a growing interest in the identification and design of new AMPs that are more potent, selective, and cost-effective.

Supporting conventional antibiotics by AMPs may enhance the effectiveness of antimicrobial treatment and reduce the problem of increasing resistance, making combination therapies a promising avenue for future research.

AMPs can also be used as an alternative to conventional antibiotics in animal feed to prevent and treat bacterial infections. This could help reduce the use of antibiotics in agriculture, which has contributed to the development of antibiotic-resistant bacteria.

Some AMPs, such as LL-37 or defensins, can work by direct killing, which involves disrupting the microbial cell membrane, leading to cell lysis and death. AMPs can also work by modulating the immune response. For example, LL-37, in addition to its direct killing activity, can also modulate the immune response by attracting immune cells to the site of infection and promoting the release of inflammatory cytokines.

AMPs, due to their cationic properties, bind preferentially to cancer cells, causing the disruption of the lipid membranes. Activating various intracellular signalling pathways leads to apoptosis and cell death. These properties make AMPs a very attractive target for clinical development. There are many FDA-approved AMPs that operate in ways that are often irreplaceable, e.g., in the case of antibiotic-resistant bacteria infection, such as *Staphylococcus aureus*. AMPs can be used in biomedical coatings, providing protection to the medical devices and enhancing their functionalities, such as drug delivery, implants, biosensing or osseointegration.

Cosmetology is also a promising area of AMP research. The current knowledge indicates that AMPs can provide the necessary effects to meet the needs of modern cosmetics, such as blocking ion channels, providing antioxidants, and anti-inflammatory effects, among other things. Therefore, it is likely that the cosmetic industry will continue to focus on the development of new products based on AMPs.

AMPs can be used as ingredients in skincare products such as cleansers, moisturisers, and serums. Research has shown that some AMPs have the ability to kill acne-causing bacteria and may also have anti-inflammatory properties, making them promising for the treatment of acne and other skin conditions. AMPs have also been investigated for their potential use in hair care products. Some AMPs have been shown to promote hair growth and improve the appearance of damaged hair. Some AMPs have been shown to have antioxidant properties and may be effective in protecting the skin from damage caused by UV radiation.

The global antimicrobial peptide market is poised to experience significant growth in the coming years due to the rising demand for effective and safe medicines, increased prevalence of infectious diseases, and expanding research activities.

Despite the challenges associated with the practical application of AMPs, including stability and half-life issues, the market is anticipated to benefit from the growing pool of scientists and talented technocrats interested in the arena of innovative therapeutics containing antimicrobial peptides for rare diseases.

## Figures and Tables

**Figure 1 ijms-24-09031-f001:**
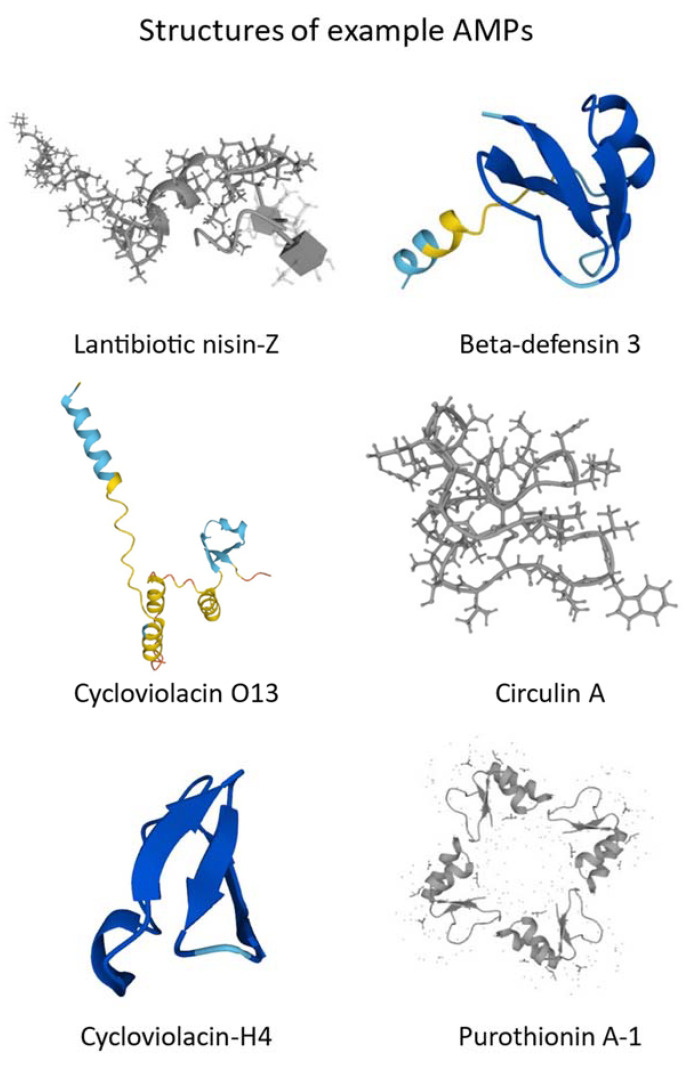
Structures of representative AMPs (created with UniProt database). Colours stand for Structure Model Confidence: Navy blue-Very high (pLDDT > 90), Blue-Confident (90 > pLDDT > 70), Yellow-Low (70 > pLDDT > 50), Orange-Very low (pLDDT < 50). pLDDT stands for predicted local distance difference test.

**Figure 2 ijms-24-09031-f002:**
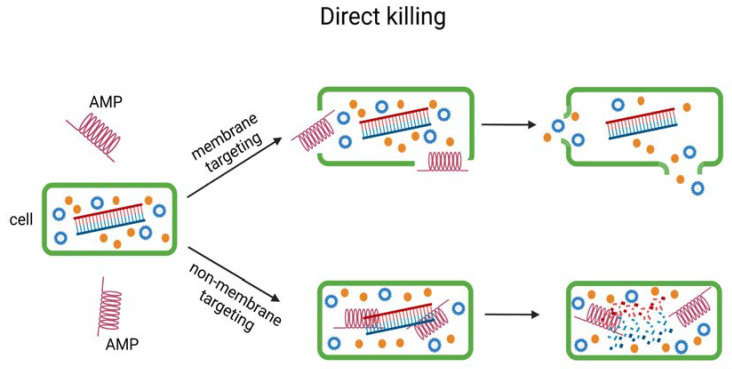
The direct killing mechanism of AMPs includes membrane targeting and non-membrane targeting. Green-bacterial membrane; Orange and Blue-cell organelles.

**Figure 3 ijms-24-09031-f003:**
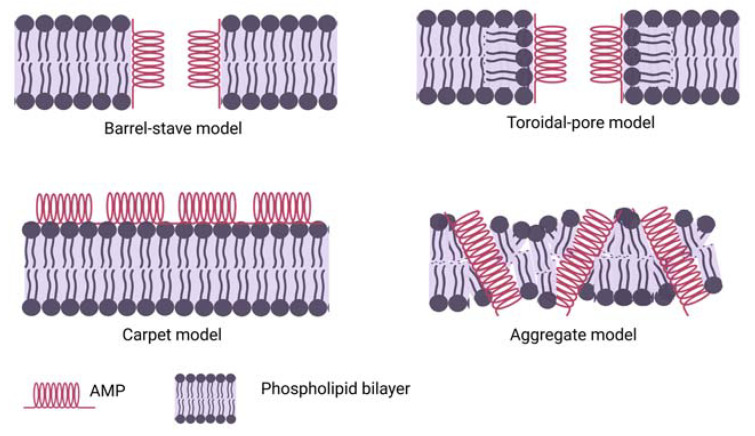
Direct killing mechanism of AMP action. There are four main types of AMP interactions with cell membranes, named barrel-stave model, toroidal-pore model, carpet model and aggregate model.

**Figure 4 ijms-24-09031-f004:**
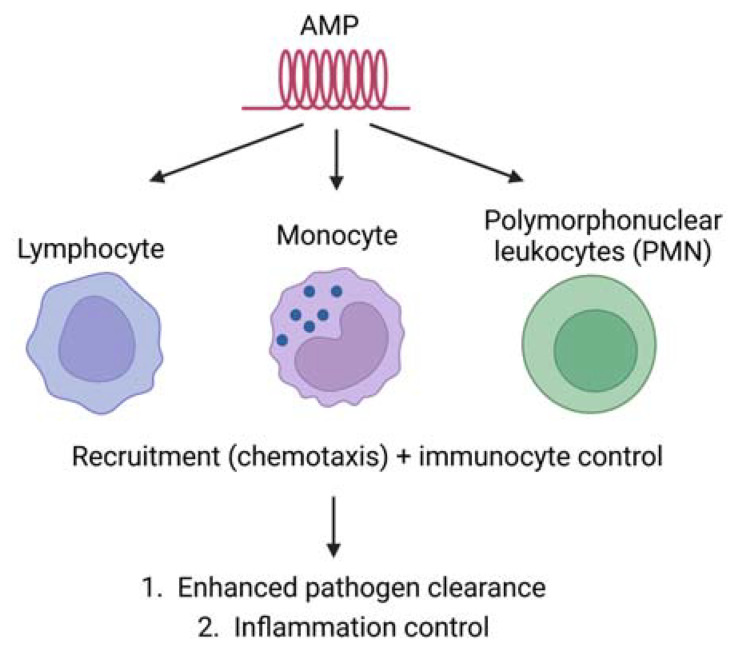
Immune modulation AMPs can recruit and activate immune cells. It results in the control of inflammation and increased cell killing.

**Figure 5 ijms-24-09031-f005:**
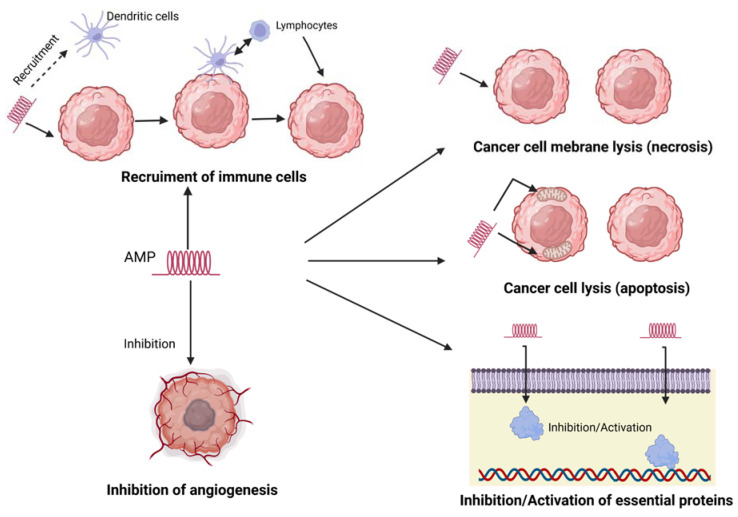
Different mechanism of AMP anticancer action.

**Figure 6 ijms-24-09031-f006:**
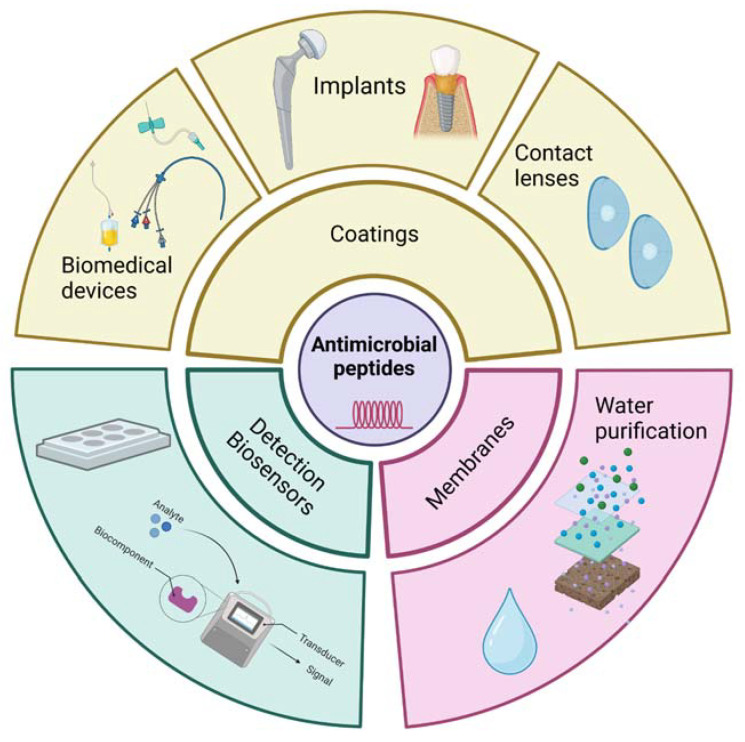
Biomedical applications of AMPs.

**Table 1 ijms-24-09031-t001:** Classes of antimicrobial peptides based on structure [29].

Cat.	AMP	Structure	Source	Ref.
α-helical peptides	Aurerin 1–2	Amidated C-terminus	Frogs	[30,31]
	Melittin	Amidated C-terminus	Bees	[32]
	Brevinin 1	-	Frogs	[33]
	Maculatins	Amidated C-terminus	Frogs	[34]
	Citropin	Amidated C-terminus	Frogs	[35]
	Buforin II	-	Toad	[36]
	Cathelicidins			[37]
	(LL-37	Amidated C-terminus	Human	
	BMAP-27,28,34	-	Bovine	
	Magainins	-	Frogs	
	Cecropin)	Amidated C-terminus	Insect	
β-sheet peptides	Cathelicidins (Protegrins Bactenecin) Defensins	Cysteine-rich Disulphide forming loop	Pigs Bovine Mammals	[37]
	Tachyplesins and Polyphemusin	Three disulphide bonds Cysteine/arginine-rich and amidated C-terminus	Horse Crab	[32,38,39,40]
			Horse Crab	[41]
Extended structure	Cathelicidins (PR-39 Tritrpticin Indolicidin Crotalicidin 15–34)	Proline and arginine-rich Tryptophan and arginine-rich Tryptophan and amidated C-terminus Lysine rich	Pigs Pigs Bovine Snakes Humans	[37]
	Histatins	Histidine-rich and amidated C-terminus		[42]

**Table 2 ijms-24-09031-t002:** List of some of the AMPs that can be used in cancer therapeutic studies [158].

AMPs	Source	Significance	Ref.
Poca A, Poca B and CyO4	*Pombalia calceolaria*	Reduced the breast cancer cell up to 80%	[185]
Aurein 1.2	*Frog Litoria aurea*	Among 54 cancer cells, 52 are inhibited in NCI testing method	[186]
Bmattacin2	*Bombyx mori*	Disrupted A375 and HCT116 cancer cells	[187]
Laterosporulin10	*Brevibacillus* sp.	MCF-7, H1299, HEK293T, HT1080, and HeLa cancer cells were disrupted	[188]
Dermaseptin-PD-1 and dermaseptin-PD-2	*Phyllomedusine leaf frogs*	Growth of H157, PC-3, and U251 MG cancer cell was inhibited	[189]
Scolopendrasin VII	*Centipede*	Reduction in the viability of leukaemia cells	[190]
Myristoyl-CM4	*Synthetic*	Activates caspase 9, caspase 3, and cleavage of Poly(ADP-ribose) polymerase (PARP) in breast cancer cells	[191]
K4R2-Nal2-S1	*Synthetic*	Binds with lung cancer cells and results in apoptosis	[192]
VLL-28	*Sulfolobus islandicus*	Inhibits murine and human tumour cells	[193]
CopA3	*Copris tripartitus*	Reduction in cell viability of gastric cancer cells	[194]
Pardaxin	*Pardachirus armoratus*	Improved the activation of caspase-3	[195]

## Data Availability

Data sharing not applicable.

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
