# Peer review of "Antimicrobial Peptides: Challenging Journey to the Pharmaceutical, Biomedical, and Cosmeceutical Use"

_ijms, 2023, doi:10.3390/ijms24109031_

Round 1

Reviewer 1 Report

The topic in question is certainly interesting and could be of great benefit to the scientific community. However, before accepting it for publication in the IJMS, several concerns should be addressed as follows:

1. Abstract: it should be improved. Line 14: please correct (uses). Line 17: please correct as follows (In this review, we would like to present the structure, mechanisms of action, possible applications, methods of production, and market for AMPs).

2. The paper should be checked for clarity and accuracy of language. It should be written in a clear and concise way, with all the relevant information presented in an easy-to-understand manner. The paper should also be checked for any spelling or grammatical errors, as these could affect the overall quality of the paper.

3. Figure 1: it is not clear. Non-membrane targeting (please explain which structure you mean.) you should add a figure, illustrating the biomedical applications of AMPs. Is there a difference of mechanism of action according to the kind of bacteria (G+ve) or G-ve? Please explain using a figure. You should incorporate a table to present the previous anticancer applications of AMPs, including type of cancer, major findings, in vivo or in vitro studies. Pharmaceutical and biomedical applications should be divided into subsection according to the discussed application.

4. Future perspectives should be added.  

5. References should be updated.

The paper should be checked for clarity and accuracy of language. It should be written in a clear and concise way, with all the relevant information presented in an easy-to-understand manner. The paper should also be checked for any spelling or grammatical errors, as these could affect the overall quality of the paper.

Author Response

Dear Editor and Reviewers,

We are pleased to submit our revised manuscript entitled  “Antimicrobial Peptides: Challenging Journey to the Pharmaceutical, Biomedical, and Cosmeceutical Use”.We thank the Editor and the Reviewers for their time to evaluate our manuscript. We have addressed all of the raised issues. In the revised manuscript, we have marked the texts in red where changes were made based on the Reviewers’ suggestions. We hope that the prepared revision of the manuscript will convince the Editor and the Reviewers about the importance of our work.

With kind regards,

Anna Mazurkiewicz-Pisarek

Reviewer #1: 

We thank the Reviewer for taking the time to review our manuscript. We believe that the Reviewer’s valuable input has made our manuscript better and it will help us communicate with the potential readers more clearly. We have addressed Reviewer’s comments in the following point-by-point answers. 

1.Abstract: it should be improved. Line 14: please correct (uses). Line 17: please correct as follows (In this review, we would like to present the structure, mechanisms of action, possible applications, methods of production, and market for AMPs). 

The abstract has been improved according to the Reviewer's comments.

  1. The paper should be checked for clarity and accuracy of language. It should be written in a clear and concise way, with all the relevant information presented in an easy-to-understand manner. The paper should also be checked for any spelling or grammatical errors, as these could affect the overall quality of the paper. 

Thanks to the reviewer for this suggestion. The article has been checked for clarity and linguistic correctness. Manuscript has also been corrected for spelling and grammatical errors.

  1. Figure 1: it is not clear. Non-membrane targeting (please explain which structure you mean.) 

Figure 1 has been modified according to the Reviewer suggestion.  

 Is there a difference of mechanism of action according to the kind of bacteria (G+ve) or G-ve?

The difference in mechanism of action according to the kind of bacteria (G+) or (G-) has been described.

You should add a figure, illustrating the biomedical applications of AMPs. Please explain using a figure. 

The figure illustrating the biomedical applications of AMPs has been added.

You should incorporate a table to present the previous anticancer applications of AMPs, including type of cancer, major findings, in vivo or in vitro studies. 

The table presenting the anticancer properties of AMPs has been incorporated in the manuscript.

Pharmaceutical and biomedical applications should be divided into subsections according to the discussed application. 

Pharmaceutical and biomedical applications have been divided into subsections. 

  1. Future perspectives should be added.  

The future perspectives have been added to the conclusion section.

  1. References should be updated.

All references were updated in the revised manuscript.

Reviewer 2 Report

The present review aims to present structure, mechanisms of action, possibility applications, methods of production and market of antimicrobial peptides. It involves 309 articles, with rich research content and large workload, which can provide readers with the latest progress in the field. Therefore, the article can be accepted after minor revision.

1. The content of the abstract is slightly too little, which is less attractive to readers. It is suggested to appropriately expand the content and add hot keywords and innovation points.

2. Provide the full name when AMR first appears.

3. Line 43 is missing a space.

4. The table should use a three line grid.

5. Can you provide the activity data of the representative compounds in part 1.

6. Can you also provide a schematic diagram of the anticancer mechanism in Part 2.3.

7. The article covers a wide range of knowledge and rich content, but the specific details and data are slightly general, and some data support needs to be added appropriately.

very good.

Author Response

Dear Editor and Reviewers,

We are pleased to submit our revised manuscript entitled  “Antimicrobial Peptides: Challenging Journey to the Pharmaceutical, Biomedical, and Cosmeceutical Use”.We thank the Editor and the Reviewers for their time to evaluate our manuscript. We have addressed all of the raised issues. In the revised manuscript, we have marked the texts in red where changes were made based on the Reviewers’ suggestions. We hope that the prepared revision of the manuscript will convince the Editor and the Reviewers about the importance of our work.

With kind regards,

Anna Mazurkiewicz-Pisarek

Reviewer #2: 

We thank the Reviewer for taking the time to review our manuscript. We believe that the Reviewer’s valuable input has made our manuscript better and it will help us communicate with the potential readers more clearly. We have addressed Reviewer’s comments in the following point-by-point answers. 

  1. The content of the abstract is slightly too little, which is less attractive to readers. It is suggested to appropriately expand the content and add hot keywords and innovation points.

The content of the abstract has been expanded. The keywords have been added.

  1. Provide the full name when AMR first appears.

An extended version of an acronym AMR has been provided.

  1. Line 43 is missing a space.

In line 43, a space has been added.

  1. The table should use a three line grid.

All tables have been changed.

  1. Can you provide the activity data of the representative compounds in part 1. 

 The activity data of the representative compounds was provided.

  1. Can you also provide a schematic diagram of the anticancer mechanism in Part 2.3.

The schematic diagram of the anticancer mechanism in Part 2.3 has been provided.

  1. The article covers a wide range of knowledge and rich content, but the specific details and data are slightly general, and some data support needs to be added appropriately

Specific details have been added.

Reviewer 3 Report

The topic proposed in this review is highly relevant as evidenced by the number of references and many of them are very recently published. The authors have intended to give their version of the importance of antimicrobial peptides from pharmaceutical, biomedical and finally cosmetic points of view.

The article has been structured in several parts:

General Introduction.- They describe what AMPs are and their biological importance. In this part a number of misleading expressions have been pointed out in the attached document. AMPs have been defined as an abbreviation for antimicrobial peptides, so using the expression AMP peptide is redundant.

Classification: Although they indicate that AMPs can be classified under different criteria, the authors present a table where the main types are indicated according to their structure, but then they focus on extending on AMPs derived from plants, describing five different types of peptides obtained from plants with their main actions.  In my opinion, the article lacks some other scheme that summarizes or focuses the attention on what is being indicated.

Mechanisms of action: In this section there are some diagrams that help the text.

Applications: This is the central part of the work and contains the most information. That is why I miss again a scheme or graphs that summarize all the most important information. If possible, a drawing of molecules, perhaps of the most important ones. The description of so much information makes the reader get lost among so many references.

Production methods: A vision of the current state of the subject and what will be its future is given.

Conclusions: They are adequate.

References.

I have found numerous mistakes in the description of the citations. Authors should check well how journals, authors, years etc. are written. Many of them are marked in the attached document in yellow, the red marks indicate errors that should be corrected.

This article is a review article so the literature section should be the most careful. In my opinion, the article lacks a series of diagrams or graphs to help the reader with all the information being given.

Author Response

Dear Editor and Reviewers,

We are pleased to submit our revised manuscript entitled  “Antimicrobial Peptides: Challenging Journey to the Pharmaceutical, Biomedical, and Cosmeceutical Use”.We thank the Editor and the Reviewers for their time to evaluate our manuscript. We have addressed all of the raised issues. In the revised manuscript, we have marked the texts in red where changes were made based on the Reviewers’ suggestions. We hope that the prepared revision of the manuscript will convince the Editor and the Reviewers about the importance of our work.

With kind regards,

Anna Mazurkiewicz-Pisarek

Reviewer #3: 

We thank the Reviewer for taking the time to review our manuscript. We believe that the Reviewer’s valuable input has made our manuscript better and it will help us communicate with the potential readers more clearly. We have addressed Reviewer’s comments in the following point-by-point answers. 

The topic proposed in this review is highly relevant as evidenced by the number of references and many of them are very recently published. The authors have intended to give their version of the importance of antimicrobial peptides from pharmaceutical, biomedical and finally cosmetic points of view.

The article has been structured in several parts:

General Introduction.- They describe what AMPs are and their biological importance. In this part a number of misleading expressions have been pointed out in the attached document. AMPs have been defined as an abbreviation for antimicrobial peptides, so using the expression AMP peptide is redundant.

All abbreviations of AMP have been changed in Manuscript. 

Classification: Although they indicate that AMPs can be classified under different criteria, the authors present a table where the main types are indicated according to their structure, but then they focus on extending on AMPs derived from plants, describing five different types of peptides obtained from plants with their main actions.  In my opinion, the article lacks some other scheme that summarizes or focuses the attention on what is being indicated.

Additional schemes representing the content of the manuscript have been prepared.

Mechanisms of action: In this section there are some diagrams that help the text.

Applications: This is the central part of the work and contains the most information. That is why I miss again a scheme or graphs that summarize all the most important information. If possible, a drawing of molecules, perhaps of the most important ones. The description of so much information makes the reader get lost among so many references.

Additional figures have been added to manuscript.

Production methods: A vision of the current state of the subject and what will be its future is given.

Conclusions: They are adequate.

References.

I have found numerous mistakes in the description of the citations. Authors should check well how journals, authors, years etc. are written. Many of them are marked in the attached document in yellow, the red marks indicate errors that should be corrected.

All of the errors in the references have been checked and corrected.

This article is a review article so the literature section should be the most careful. In my opinion, the article lacks a series of diagrams or graphs to help the reader with all the information being given.

The manuscript has been enriched with tables and figures that facilitate navigation through the text.

Round 2

Reviewer 1 Report

The authors did a great job handling the previous claims. They provided evidence and data to back up their arguments. Furthermore, they addressed the doubts and criticism in a comprehensive and convincing manner.

Author Response

We thank the Reviewer for taking the time to review our manuscript after major revision. We believe that the Reviewer’s valuable input has made our manuscript better and it will help us communicate with the potential readers more clearly. 

With kind regards,

Anna Mazurkiewicz-Pisarek

Reviewer 3 Report

This new version is an improvement over the previous one.

New tables and illustrative figures have been improved and introduced, although the numbers need to be corrected because the text does not correspond to the figure.

The distribution of the information has been changed, making it easier to follow.

Despite the improvements, there are things that need to be corrected and which have been marked in yellow in the attached document.

When fungi and bacteria are mentioned, they should always be written in italics.

There are very repetitive expressions in the paragraphs, so an alternative should be found for some of them.

In the bibliography new references have been included, related to the new information, the errors of the previous version have been corrected, but there are still some that have been marked in yellow.

Author Response

We thank the Reviewer for taking the time to review our manuscript once again after major revision. We have addressed Reviewer’s comments in the following point-by-point answers. 

This new version is an improvement over the previous one.

New tables and illustrative figures have been improved and introduced, although the numbers need to be corrected because the text does not correspond to the figure.

Table numbers have been corrected to reflect the texts in the manuscript. 

The distribution of the information has been changed, making it easier to follow.

Thank you for confirmation that our changes made the Manuscript easier to follow.

Despite the improvements, there are things that need to be corrected and which have been marked in yellow in the attached document.

All items have been corrected and marked in blue in the Manuscript based on the Reviewers’ suggestions. 

When fungi and bacteria are mentioned, they should always be written in italics.

The names of fungi and bacteria were written in italics.

There are very repetitive expressions in the paragraphs, so an alternative should be found for some of them.

The repetitive expressions in the paragraphs have been corrected.

In the bibliography new references have been included, related to the new information, the errors of the previous version have been corrected, but there are still some that have been marked in yellow.

The errors in the bibliography have been corrected.

With kind regards,

Anna Mazurkiewicz-Pisarek
